# Full-Scale Measurements of Wind Characteristics on a High-Rise Building during Typhoon Sarika

**Jiaxing Hu [1,*], Zhengnong Li [2] and Zhefei Zhao [3]**

1   School of Civil and Environmental Engineering, Hunan University of Science and Engineering, Yongzhou 425199, China

2   Key Laboratory of Building Safety and Energy Efficiency of the Ministry of Education, Hunan University, Changsha 410082, China; zhn88@263.net

3   School of Vocational Engineering, Health and Sciences, Royal Melbourne Institute of Technology, GPO Box 2476, Melbourne, VIC 3001, Australia; zhefeizhao@gmail.com

*   Correspondence: wjxcivil@yeah.net

**Abstract:** A full-scale measurement of wind characteristics atop a high-rise building (with a height of 115 m) was conducted during the passage of Typhoon Sarika on 18 October 2016. Wind field characteristics, wind speed, and wind direction atop the building were recorded synchronously, and turbulence intensity, turbulence integral scale, gust factor, and power spectrum were investigated. Meanwhile, the time and frequency domain characteristics of the wind field were analyzed. The stationarity test results of Typhoon Sarika at different time steps are researched in a runs test. And the time-frequency analysis of non-stationary samples of fluctuating wind speed are conducted by wavelet transform, the measured data are valuable for the wind-resistant design of high-rise buildings in typhoon-prone regions.

**Keywords:** full-scale measurement; typhoon; wind characteristics; turbulence intensity; turbulence integral scale; gust factor

## 1. Introduction

With the rapid development of data acquisition system and sensor technology, field measurement of wind characteristics on high-rise buildings under typhoon is an enduring and developing research topic. Lots of field measurements of wind characteristics of high-rise buildings during typhoons have been conducted. Since the 1990s, many scholars, represented by Professor Li Q.S., have carried out massive field measurements on more than 10 super high-rise buildings in coastal areas of China [1–4]. The measurements covered the wind field atop the high-rise buildings, the characteristics of internal and external pressure on the building surface, the vibration displacement and acceleration response of high-rise buildings, etc. These field measurements provide valuable references for enhancing the understanding of the wind characteristics and structural dynamic characteristics (vibration mode, natural frequency, and damping ratio) of high-rise buildings. Xu et al. [5] installed the wind and structural monitoring system on Di Wang Tower under Typhoon York to monitor wind and structural response data. The field measurement data were analyzed clearly and demonstrated that the performance of Di Wang Tower is satisfactory during Typhoon York, which was a valuable finding for further wind-resistant design. Li L.X. et al. [6] conducted field measurements in typhoon Hagupit to model the spectral features of typhoon winds with critical implications on the mitigation of disproportionate damage experienced in typhoon-prone regions, and proposed a data-driven model for the wind power spectrum in tropical cyclone winds over the sea surface. Cao et al. [7] studied the gust factor related to Typhoon Maemi and showed that typhoon gust factors have no significant difference from those associated with non-typhoon winds, and can be described using models developed for standard neutral boundary layer flow conditions. He et al. [8]

investigated the gust factors during Super Typhoon Hato in the South China sea, the results show that the gust factor initially increased with increasing wind speed and plateaued at wind speed between 30 and 50 m/s before exhibiting a decreasing trend until 70 m/s. Fang et al. [9] investigated the gust characteristics of near-ground typhoon based on 14 sets of records observed by 4 meteorological stations during 10 typhoons, the results show that the off-sea gust factor at 10 m is 8% higher than the over-land value while other heights show little difference. Lin et al. [10] conducted field measurements of two Typhoon cyclones to study the characteristics of the near-surface boundary layer, and found a region-specific wind speed power spectrum and its associated spectrum parameters that can be used for the design of wind-resistant structures in Typhoon-prone regions. Li et al. [11] analyzed the field measurement data of wind-excited vibration responses of two skyscrapers (393 and 432 m in height) during Super Typhoon Mangkhut. Li Z.N. et al. [12] conducted many field measurements of dynamic responses and wind-filed characteristics on 5 high-rise buildings located in the coastal regions of Wenzhou, Xiamen and Haikou from 2008 to 2018, by which they investigated wind field characteristics of different geomorphic characteristics under typhoon and the time-domain and frequency-domain characteristics of the fluctuating wind speed acceleration response.

The southeast coast of China is a typhoon-prone region, the measured high-rise building located in the coastal area of Haikou City was thus selected to monitor more typhoons considering that Hainan Island has the maximum probability of typhoon landing. The field measurement of wind characteristics atop high-rise buildings under typhoon is far from enough, especially the measured samples of wind characteristics atop super high-rise buildings under the action of strong typhoons. The geographical coordinate of measured building along the north axis is 11° by West, and the basic wind pressures of 10-year, 50-year, and 100-year return periods are 0.45 kN/m$^2$, 0.75 kN/m$^2$, and 0.90 kN/m$^2$ respectively considering the high standard of wind-resistant design of buildings in Haikou city. Based on the field measured data of the wind field on the roof under the influence of five typhoons, the characteristics of average wind and fluctuating wind during typhoon landing are analyzed, mainly including turbulence intensity, turbulence integral scale, gust factor, probability density function, and fluctuating wind speed spectrum, as well as the internal relationship between these parameters and mean wind speed and wind direction. This paper reveals the relationship between gust factor and the basic time interval and gust duration, puts forward the empirical formula for predicting the relationship between gust factor and turbulence intensity, and discusses the characteristics of fluctuating wind speed spectrum at different stages of typhoon landing. The measured building was elected to conduct wind-resistant field measurement and analyze the impact of different inflow wind field types on the wind-induced response of high-rise buildings due to the diverse geomorphic characteristics of the building in different directions.

## 2. Full-Scale Measurement Program

### 2.1. Typhoon Sarika

As reported by China Meteorological Administration (http://www.cma.gov.cn/), Typhoon Sarika formed as a tropical depression over the western North Pacific Ocean to the east of the Philippines on 12 October 2016, and landed on the eastern coast of Luzon Island, Philippines at 2:20 on 16 October 2016, reaching its peak intensity with the maximum magnitude wind of 16 (55 m/s) near its centre and with a minimum air pressure of 930 hPa. After moving westward rapidly across the northern part of the South China Sea for two days, Sarika made landfall along the coast of Hele Town of Wanning city, Hainan Province at 9:50 on 18 October 2016, with a maximum magnitude wind of 14 (45 m/s) near its centre and with a minimum air pressure of 955 hPa. The typhoon Sarika, which decayed rapidly on 19 October at 17:00, made its final landfall over the coastline and border of Vietnam and China. The moving track of Sarika and the ambient condition of the experimental site is presented in Figure 1.

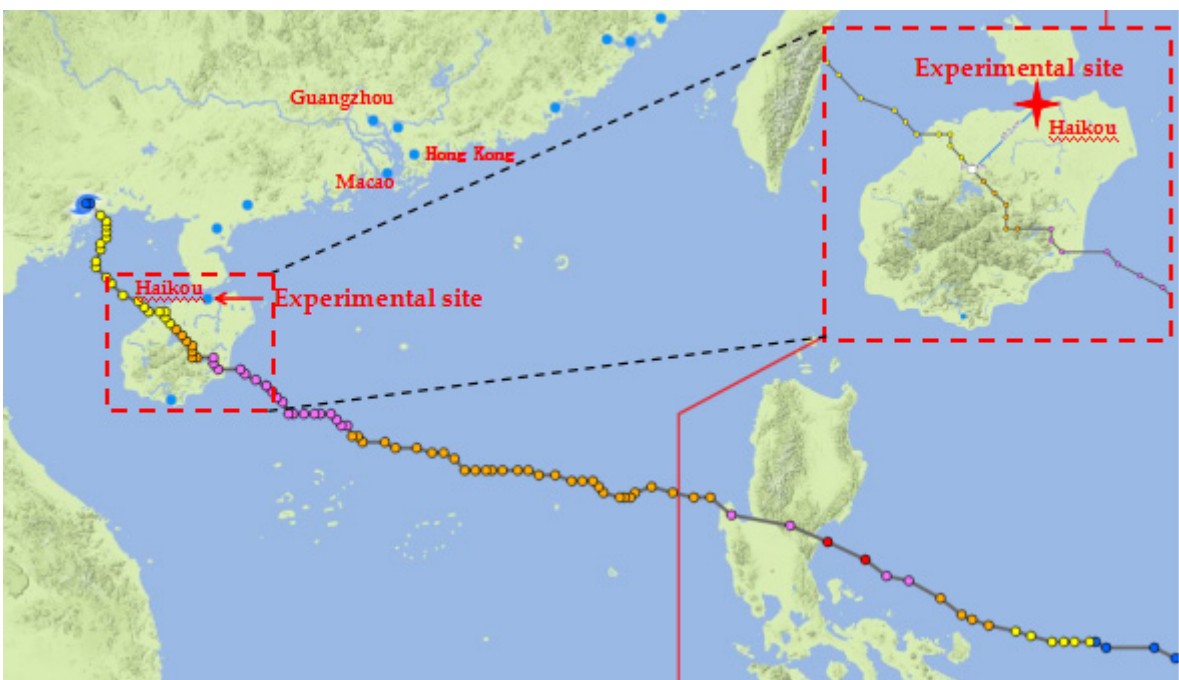

**Figure 1.** Track of typhoon Sarika and the ambient condition of experimental site.

### 2.2. The Ambient Condition of Experimental Site

The measured experimental site is on the top floor of a high-rise building supported by four massive columns near the coast of the South China Sea. The measured building lies in the typhoon-prone region with a latitude of 20°02′ N and a longitude of 110°35′ E. The building is situated on the north coast of Haikou and is approximately 1.2 km from the shoreline. There are several low-rise buildings which are lower than 80 m around the measured building. Figure 2 illustrates the surrounding building environment near the experimental site. The terrain roughness in Chinese codes [13] is classified into four categories: A, B, C, and D, while it is grouped into five categories: 0, I, II, III, and IV in prEN 1991-1-4 (Eurocode, 2005). Category-A in Chinese Code corresponds to category-0 and I in the European Code, and Category-B, Category-C and Category-D correspond to the Category-II, Category-III and Category-IV, respectively. Considering that the Surrounding high-rise buildings are along specific wind directions of 30°, 190°, and 260°, so the wind-field on the roof of the measured building will be susceptible to the severe interference, which is consistent with the category D of terrain roughness specified in the Chinese code. There are numerous low-rise buildings along wind directions of 100°~160°, which is also consistent with the categories B or C of terrain roughness specified in the Chinese code. Because the wind direction of 315° is near the opening area in northwest of the sea, it corresponds to the category A of terrain roughness specified in the Chinese code. Considering that there are massive middle and high-rise buildings in China, it is more reasonable to classify the categories and types of the terrain roughness according to the Chinese codes.

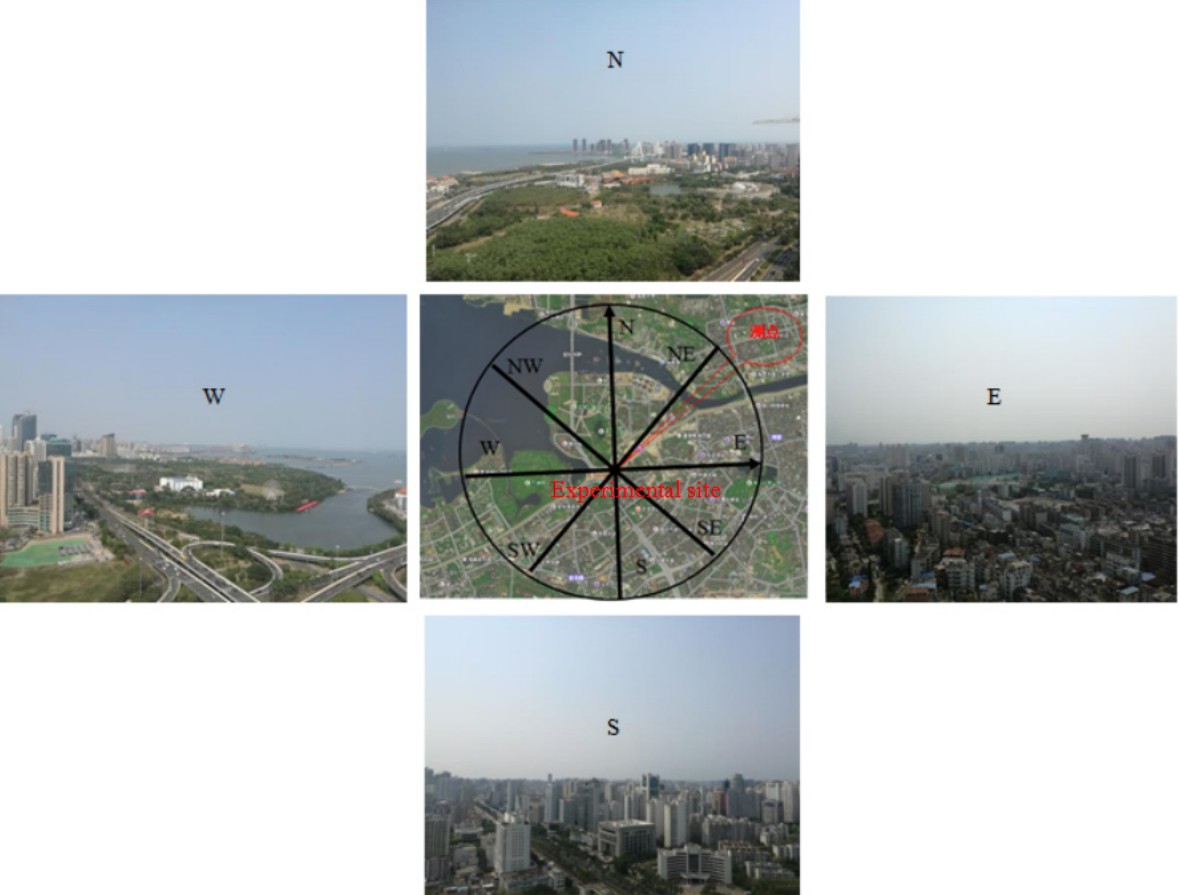

**Figure 2.** The surrounding building environment near the experimental site.

*2.3. Introduction of Measuring Instruments and Measuring Point Scheme*

The time histories of wind speed and wind direction are all collected by the monitoring system (Model uT33) produced by Wuhan uTekL Electronic Technology Co., Ltd. (Wuhan, China) with a sampling frequency of 128 Hz. Figure 3 shows the installation location of Model 05103V anemometer at a height of 115 m, the instantaneous wind speed is recorded by the mechanical anemometer with the maximum measurement capacity of 100.00 m/s. Figure 4 shows the plan of installation position of 05103V anemometer on the roof of high-rise building, the 0 direction of the anemometer is parallel to the *y*-axis and refers to North by West 11°, and the 90° of the anemometer represents the measured wind direction parallel to the *x*-axis, and the designation of other angles follows the clockwise rule.

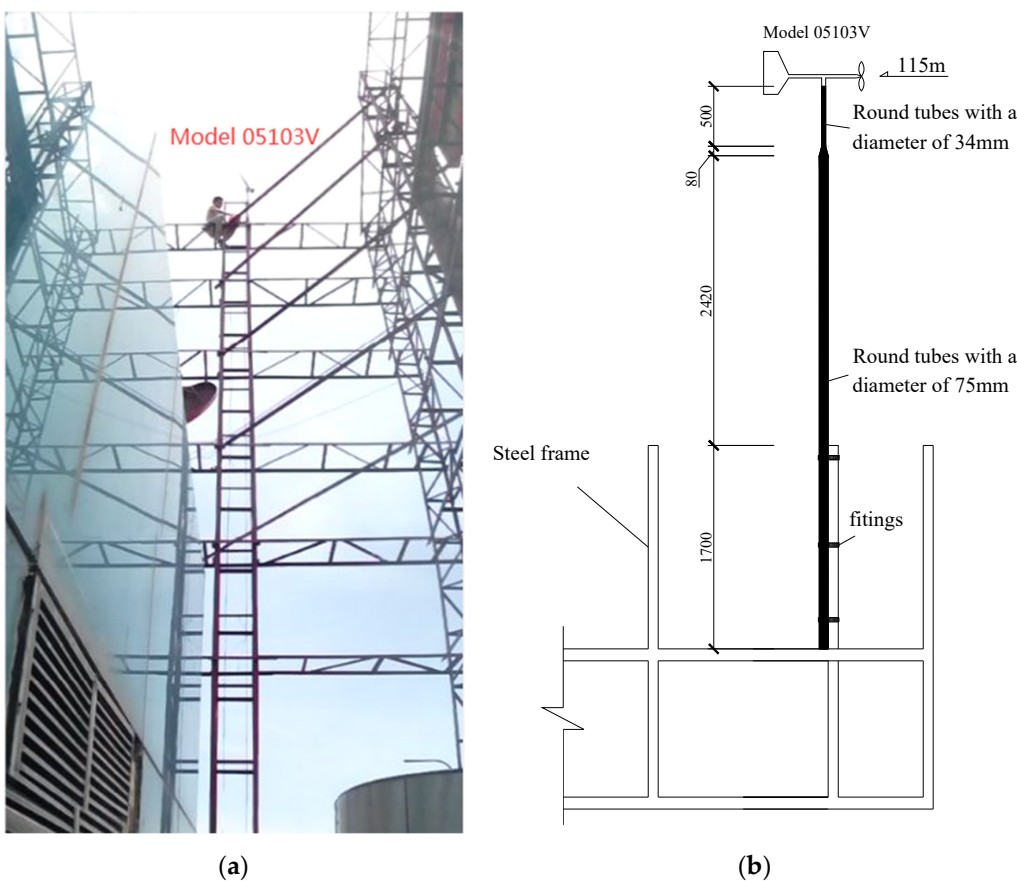

**Figure 3.** Installation location of Model 05103V anemometer. (**a**) Installation drawing of anemometer; (**b**) Installation detail drawing.

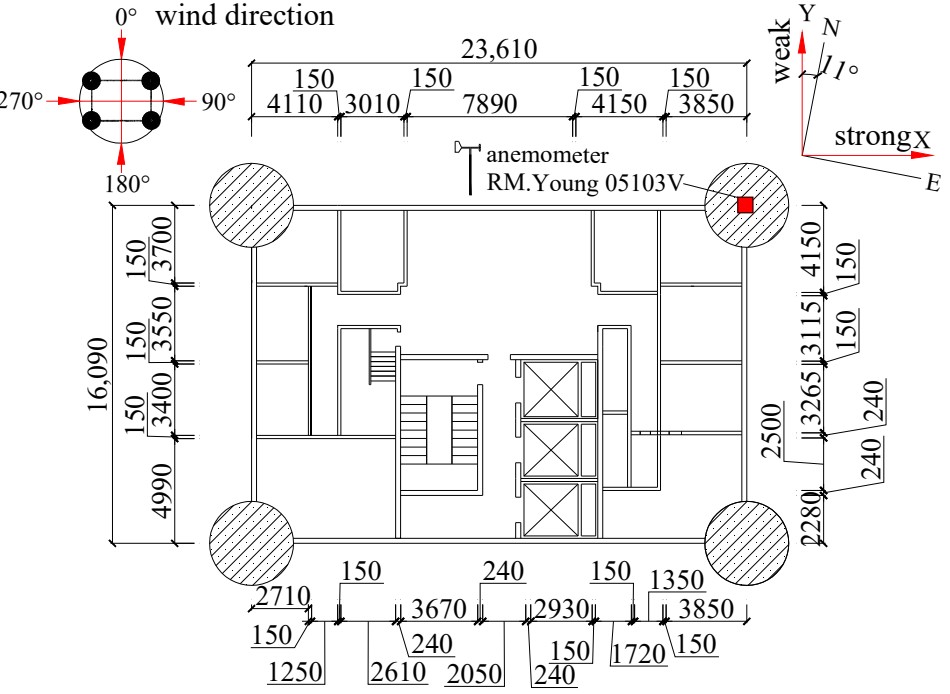

**Figure 4.** Plan of installation position of 05103V anemometer on the roof of high-rise building.

### 3. Wind Characteristics of Typhoon Sarika

Firstly, in order to prevent high-frequency signal interference in the process of typhoon wind field data acquisition, it is necessary to carry out band-pass filtering on the measured wind field data before analyzing the data. Because the frequency domain distribution of wind power spectrum is concentrated in the range of 0~12 Hz, the band-pass filtering range used in this paper is 0~12 Hz. Secondly, the measured process is often accompanied by power frequency, frequency doubling, and irregular random interference noise signals, resulting in obvious burr points in the measured data. A large peak data occasionally appears in the time history data. If it is not removed in time, it will have a great impact on the mean value and mean square deviation of the data. Therefore, after band-pass filtering the measured wind field data, the data with burr points are interpolated and replaced by the five-point cubic method to eliminate the error caused by burr points.

#### 3.1. Mean Wind Speeds and Wind Directions

In this paper, the wind characteristic data collected by the mechanical anemometer are decomposed by the vector decomposition method, and the horizontal wind speed $U_b$ and direction $\alpha(t)$ are decomposed to the orthogonal $N$ and $E$ coordinate axes directions as shown in Figure 5, and the decomposition expression is as follows:

$$\begin{cases} U_N(t) = U_H(t)\cos\alpha(t) \\ U_E(t) = U_H(t)\sin\alpha(t) \end{cases} \tag{1}$$

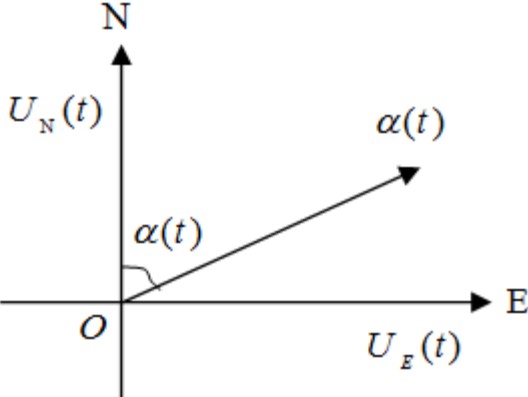

**Figure 5.** Diagram of horizontal wind speed decomposition.

The 10-min main wind speed $\overline{U}$ and main wind direction $\overline{\alpha}$ can be expressed by:

$$\begin{cases} \overline{U} = \sqrt{\overline{U}_N^2 + \overline{U}_E^2} \\ \cos\overline{\alpha} = \overline{U}_N/\overline{U} \end{cases} \tag{2}$$

where $\overline{U}_N = \frac{1}{T}\int_0^T U_N(t)dt$ and $\overline{U}_E = \frac{1}{T}\int_0^T U_E(t)dt$ are the average values of wind speed component in the orthogonal two coordinate direction within 10 min. Therefore, the formula of component calculation of along-wind wind speed and across-wind fluctuating wind is as follows:

$$\begin{cases} U(t) = U_N(t)\cos\overline{\alpha} + U_E(t)\sin\overline{\alpha} \\ v(t) = -U_N(t)\sin\overline{\alpha} + U_E(t)\cos\overline{\alpha} \end{cases} \tag{3}$$

The 36.7 h wind field data collected at 23:00 on 17 October 2016 and is used to record the wind speed and direction during the passage of Typhoon Sarika. The measured maximum instantaneous wind speed is 34.84 m/s. Before the landing of Typhoon Sarika, the wind direction varies greatly and appears to be unstable, while it shifts about 180° during the passage of Typhoon Sarika, namely the wind direction shifts almost from south to north. During the most muscular landing passage of Typhoon Sarika, the wind direction

varies in the range of 70~110°. Figure 6 shows the 10-min mean wind speed and direction variations during the passage of Typhoon Sarika, the peak of 10-min mean wind speed is about 20.02 m/s and the 10-min mean wind direction is from 0° to 180°.

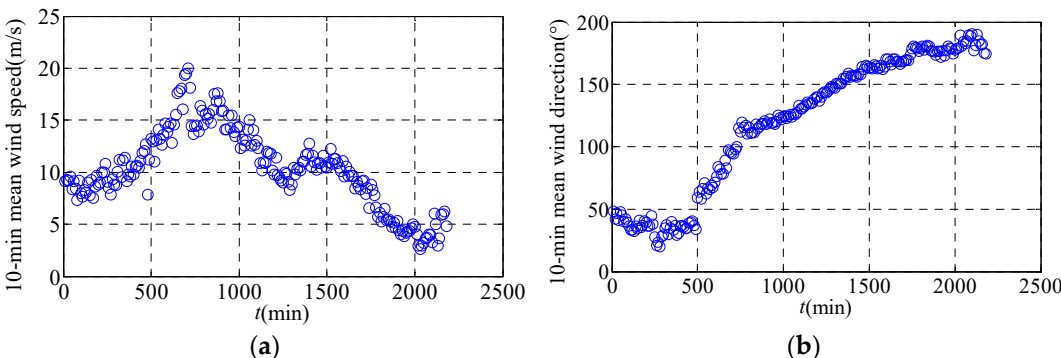

**Figure 6.** 10-min mean wind speed and direction during Typhoon Sarika. (**a**) 10-min mean wind speed (**b**) 10-min mean wind direction.

### 3.2. Typhoon Stationarity Test

A runs test is adopted to process the 36.7 h wind-field data of Typhoon Sarika collected at 23:00 on 17 October 2016, and the total wind-field samples are 220 within 10 min as presented in Figure 6. First, samples of the measured fluctuating wind speed are divided into $N$ intervals to calculate the root mean square (RMS) $\sigma_i^2$ and the average value $\sigma_0^2$ of these RMS. Then the RMS of each interval is compared with its average value alternatively, and when the RMS of the $i$th interval is greater than the average value, the section is marked as "+", otherwise, it is marked as "−". These marks are then arranged successively according to the order of each interval, and the total number of rounds is recorded as $r$ in the condition that each change of marks is defined as one round. Significance level $\alpha$ is 0.05, and if the number of recorded rounds is within the probability interval ($\alpha/2$, $1 - \alpha/2$), then the assumption under significance level $\alpha$ is accepted. Normal distribution can be used to determine the acceptance domain of the stationarity test, and the acceptance domain is the excepted number of rounds $\pm 1.96 \times$ the standard deviation of round number. If the number of rounds is within this acceptance domain, then the sample is stationary, otherwise it is non-stationary. When the interval number $N_1$ (marked as +) or the interval number $N_2$ (marked as −) is greater than 15, it can be considered that the sample size is large, thus the acceptance and negation domains of the test can be determined by the normal distribution table as follows:

$$\mu = \frac{2N_1 N_2}{N} + 1 \tag{4}$$

$$\sigma = \sqrt{\frac{2N_1 N_2 (2N_1 N_2 - N)}{N^2 (N - 1)}} \tag{5}$$

$$Z = \frac{r - \mu}{\sigma} \tag{6}$$

where $N_1$ is the number of intervals marked as "+", and $N_2$ is the number of intervals marked as "−"; $N$ is the equal subinterval, and $N = N_1 + N_2$; $r$ is the total number of rounds. When the significance level $\alpha$ is 0.05 and $|Z| \leq 1.96$ in the runs test, then the sample is stationary. The non-stationarity samples of Typhoon Sarika are investigated at different time scales; thus, the 10-min fluctuating wind speed are segmented with 30 s, 20 s, 10 s and 3 s as time steps in sequence, and the stationarity test results of 220 samples within 10 min during Typhoon Sarika are obtained. More details are shown in Table 1.

**Table 1.** The stationarity test results of Typhoon Sarika at different time steps.

| Time Steps | Stationary Samples | Non-Stationary Samples | The Ratio of Non-Stationary Samples |
|---|---|---|---|
| 30 s | 198 | 22 | 10.00% |
| 20 s | 190 | 30 | 13.64% |
| 10 s | 172 | 48 | 21.82% |
| 3 s | 113 | 107 | 51.36% |

Table 1 shows that 113 samples with 3 s as a time step in 113 samples conform to the runs test with a significance level of $\alpha = 0.05$. The non-stationary samples with a time step of 3 s account for 51.36% of the total samples, which is significantly greater than 21.82%, 13.64%, and 10.00% of the samples with 10 s, 20 s, and 30 s as time steps in sequence, indicating that the shorter the time steps, the stricter the stationarity test, and the more obvious the non-stationarity samples of Typhoon Sarika. Therefore, the non-stationary samples are eliminated based on the test results of samples with 3 s as a time step, and the wind characteristics of 113 stationary samples during Typhoon Sarika are investigated.

*3.3. Turbulence Intensity and Integral Scale*

3.3.1. Turbulence Intensity

Turbulence intensity ($I_i$, $i = u, v$) is defined as the ratio between the standard deviation of fluctuating wind ($\sigma_i$) and mean wind speed ($\overline{U}$) for a given sample duration. Figure 7 shows the relationship of the 10-min variations longitudinal and lateral turbulence intensities ($I_u$ and $I_v$) with respect to mean wind speeds and directions ($\alpha$). The measured results show that the 10-min longitudinal and lateral turbulence intensities decrease with the increasing mean wind speed, and the reduction rate of the $I_u$ and $I_v$ drops after the 10-min mean wind speed reaches an absolute value ($\overline{U} = 10$ m/s). The measured 10-min longitudinal and lateral turbulence intensities $I_u$ and $I_v$ are obviously correlated with wind direction. It is notable that $I_u$ and $I_v$ reach larger values along the 10-min wind directions of 30° and 190°, while inclining toward smaller values along the 10-min wind directions within 100°~160°. There are numerous mid-high-rise buildings located in the wind direction of 30° and 190°, and the peak turbulence intensity appears along those wind directions due to the severe interference to the wind field. And there are numerous low-rise buildings along the wind direction of 100°~160°; thus, the lowest turbulence intensity appears along those wind directions due to slight interference. According to the calculation formula of turbulence intensity along height distribution in Chinese code (GB50009-2012), the formula can be expressed as:

$$I_z(z) = I_{10}\left(\frac{z}{10}\right)^{-\alpha} \tag{7}$$

where $\alpha$ is terrain roughness parameters index, the parameters $\alpha$ corresponding to categories A, B, C, and D can be prescribed as 0.12, 0.15, 0.22, and 0.30, respectively. The value $z$ is the erection height of anemometer. The value $I_{10}$ is turbulence intensity at the height of 10 m corresponding to categories A, B, C, and D of terrain roughness, which can be prescribed as 0.12, 0.14, 0.23, and 0.39, respectively. Equation (7) can be used to calculate the turbulence intensity at a height of 115 m with respect to four categories of terrain roughness are 8.95%, 9.71%, 13.4%, and 18.8%, respectively. The measured turbulence intensity with a 10-min interval is 15–40% during Typhoon Sarika, which indicates that the measured wind field characteristic is greatly subject to the interference of the ambient condition, thus the category of terrain roughness for most of the inflow wind is D. The measured turbulence intensity is 10% along the critical wind directions of 160° is close to category A of terrain roughness.

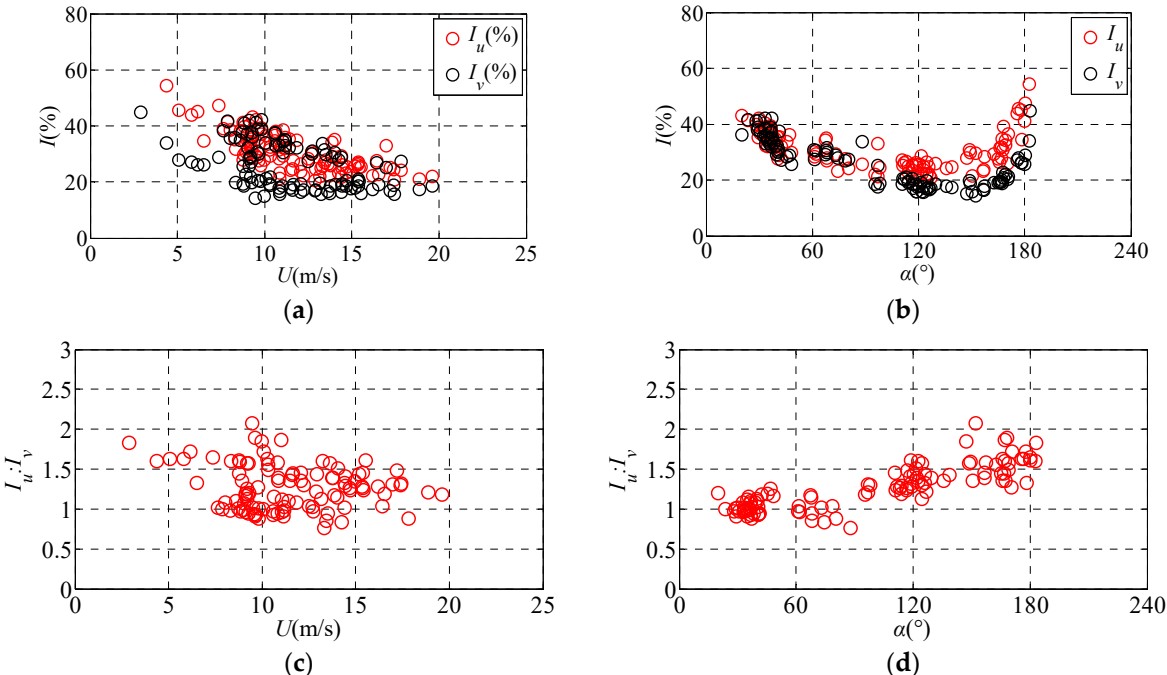

**Figure 7.** The measured longitudinal and lateral turbulence intensities. (**a**) $I_u$ and $I_v$ with respect to 10-min $\overline{U}$; (**b**) $I_u$ and $I_v$ with respect to wind direction; (**c**) $I_u$:$I_v$ with respect to 10-min $\overline{U}$; (**d**) $I_u$:$I_v$ with respect to wind direction.

In summary, the turbulence intensity of fluctuating wind closely relates to the ambient condition disturbance and terrain roughness categories. The measured turbulence intensity in the inflow-wind area is the lowest when terrain roughness is category A. Turbulence intensity decreases with the increasing mean wind speed in the same terrain roughness categories, and the longitudinal turbulence intensity are generally more significant than lateral turbulence intensities. Figure 7c shows that the ratios of the standard deviations between the turbulence components ($I_u$:$I_v$) obtained by the anemometers are relatively stable when the 10-min mean wind speed is higher than 10 m/s, and the averaged value of ratio between turbulence components ($I_u$:$I_v$) is 1.34 during the passage of Typhoon Sarika.

### 3.3.2. Turbulence Integral Scale

The Turbulence integral scale ($L_i^x$) is a measure of the average size of eddies in the turbulent flow. The values $L_u^x$ and $L_v^x$ can be expressed as longitudinal and lateral turbulence integral scales. According to Taylor's hypothesis, the turbulence integral scale can be calculated directly according to the following formula:

$$L_i^x = \frac{\overline{U}}{\sigma_i^2} \int_0^{R_i=0.05} R_i(\tau)d\tau \quad i = u,v \tag{8}$$

where $\overline{U}$ is mean wind speed; $\sigma_i$ and $R_i(\tau)$ are the standard deviation and auto-correlation function of the longitudinal and lateral fluctuating velocity components $u$ and $v$, respectively. To get a more consistent estimate of the integral scale, the auto-correlation function is integrated until the correlation drops to 0.05 and Taylor's hypothesis is used to convert the integral time scale to the integral length scale. Equation (8) shows that the turbulence integral scale is closely related to mean wind speed and the standard deviation of S.D of fluctuating wind. Therefore, the rules of turbulence integral scale concerning the mean wind speed and S.D of fluctuating wind are significant. Figure 8a,b shows the variations of the measured longitudinal and lateral turbulence integral scales with respect to the mean wind speeds and S.D of fluctuating wind with a 10-min interval. The results show that the measured turbulence integral scale is basically in the range of 0–200 m when

the 10 min mean wind speed is less than 8 m/s, while the measured turbulence integral scale is basically in the range of 200–500 m when the 10 min mean wind speed is within 8–21 m/s, Figure 8a,b shows that the measured longitudinal and lateral turbulence integral scales increase with the increasing mean wind speed and S.D of fluctuating wind, which shows that the turbulence integral scales are positively correlated with the intensity of inflow wind.

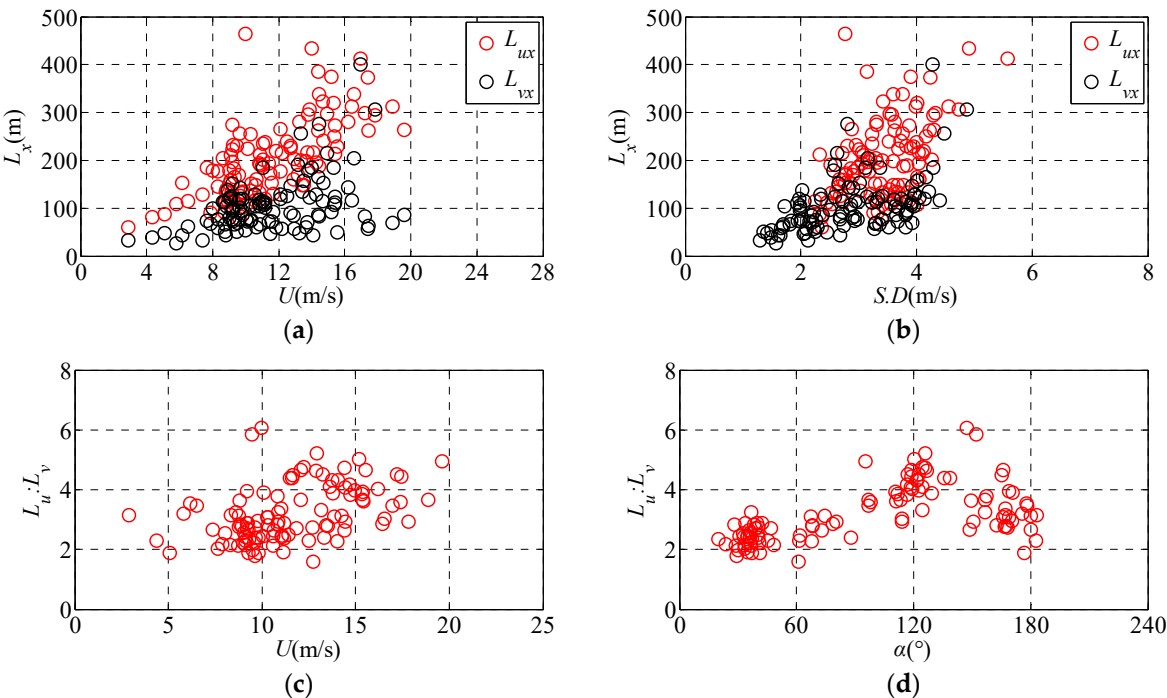

**Figure 8.** The measured longitudinal and lateral turbulence integral scales. (**a**) $L_u^x$ and $L_v^x$ with respect to 10-min $\overline{U}$; (**b**) $L_u^x$ and $L_v^x$ with respect to S.D of fluctuating wind; (**c**) $L_u^x{:}L_v^x$ with respect to 10-min $\overline{U}$; (**d**) $L_u^x{:}L_v^x$ with respect to wind direction.

Figure 8c,d shows the ratios of the turbulence scale between the turbulence components ($L_u^x{:}L_v^x$) with respect to the mean wind speed and wind direction, $L_u^x{:}L_v^x$ are basically within 2~4 under the excitation of static wind, and $L_u^x{:}L_v^x$ are within of 4~7 if the 10 min wind speeds are higher than 10 m/s, which explains that $L_u^x{:}L_v^x$ increases with the increasing mean wind speed, and further discloses that the increasing mean wind speed has greater influence on longitudinal turbulence integral scales than that on lateral turbulence integral scales.

Moreover, the ratios of the turbulence scale between the turbulence components ($L_u^x{:}L_v^x$) near 160° (close to terrain roughness category A) of the wind direction is generally larger than that along other wind direction. The results demonstrate that the ratio of the turbulence scale between the turbulence components are closely related to the terrain categories, and $L_u^x{:}L_v^x$ will reach larger values when the terrain roughness is category A.

### 3.4. Gust Factor

The gust factor $G(t_g)$ is defined as the ratio of the gust speed $\hat{U}$ in gust duration $t_g$ to the 10-min mean wind speed $\overline{U}$, which can be written as:

$$G(t_g) = \frac{\hat{U}(t_g)}{\overline{U}} = 1 + \frac{\max(\overline{u(t_g)})}{\overline{U}} \tag{9}$$

where the gust speed $\hat{U}$ is the largest mean wind speed within the gust duration $t_g$. $\max(\overline{u(t_g)})$ is the largest mean wind speed in duration $t_g$ of the longitudinal fluctuating wind speed within 10-min. It can seem that the longer the gust duration $t_g$ is, the

smaller difference between the gust duration and the gust duration is, and the results of the gust factor are closer to 1. Taking $t_g = 3$ s and $T = 10$ min as examples, this paper investigates the variations of the longitudinal and lateral gust factors concerning mean wind speed. Figure 9 shows that the longitudinal and lateral gust factors are prone to decrease when the 10-min mean wind speed is less than 7 m/s, and the longitudinal and lateral gust factors are basically (1.2, 3.2) and (0.2, 1.5) in sequence. Moreover, the longitudinal and lateral gust factors remain stable after the 10-min mean wind speed reaches 7 m/s; the longitudinal and lateral gust factors are basically (1.2, 1.8) and (0.2, 0.6) in sequence during the passage of Typhoon Sarika.

The gust factor is obviously correlated with turbulence intensity. Ishizaki [14] suggest the following formula to establish the relationship between gust factor and turbulence intensity,

$$G_u(t_g) = 1 + k_1 I_u^{k_2} \ln(T/t_g) \qquad (10)$$

where $T$ is a different sample duration. Ishizaki suggested $k_1 = 0.5$, $k_2 = 1.0$; Equation (10) shows that the main factors of gust factors are turbulence intensity, sample duration, and gust duration. Taking $t_g = 3$ s and $T = 10$ min as example, Equation (10) is adopted to fit the relationship between gust factor and turbulence intensity during the passage of Typhoon Sarika, and to compare with the linear model proposed by Ishizaki [14] Figure 10 demonstrates that the gust factor increases with the increasing turbulence intensity, and shows a linear relationship between turbulence intensity and gust factor. The results show that the prediction values of the linear model proposed by Ishizaki [14] are larger than the measured values. Based on the measured values of longitudinal and lateral gust factors and turbulence intensity during Typhoon Sarika, an empirical formula concerning longitudinal and lateral gust factor and turbulence intensity are proposed:

$$G_u(t_g) = 1 + 0.42 I_u^{0.97} \ln(T/t_g) \qquad (11)$$

$$G_v(t_g) = 0.40 I_v^{0.95} \ln(T/t_g) \qquad (12)$$

This study investigates the variation of gust factor during the passage of Typhoon Sarika obtained for different sample duration $T$ (1, 10, 30, and 60 min) and gust duration $t_g$ (0.1, 0.2, 0.3, 0.5, 1.0, 2.0, 3.0, 5.0, 10, 30, 60, 120, 180, 600, 1200, and 1800 s). Figure 11a,b shows that, the gust factor decreases as the gust duration $t_g$ increases when the sample duration $T$ is 10-min during different wind speeds. Figure 11c,d illustrates that the gust factor increases with the increasing sample duration $T$.

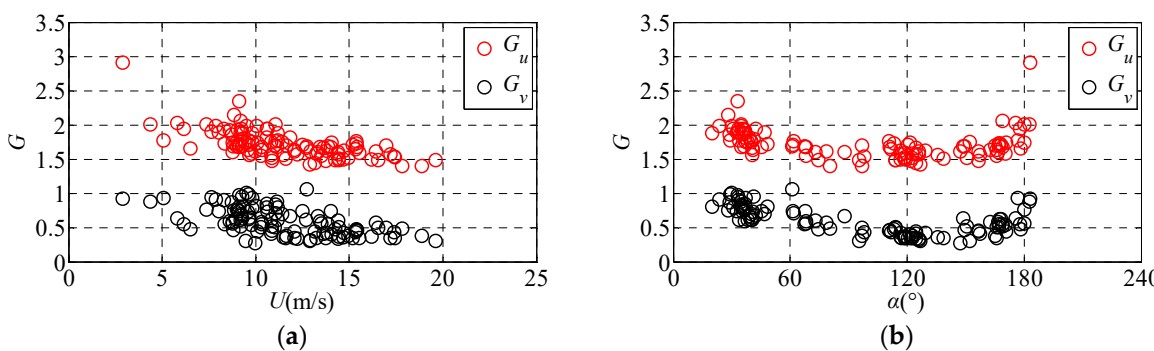

**Figure 9.** The variation of gust factor with respect to 10-min wind speed and direction. (**a**) $G_{u,v}$ with respect to 10-min $\overline{U}$; (**b**) $G_{u,v}$ with respect to 10-min $\alpha$.

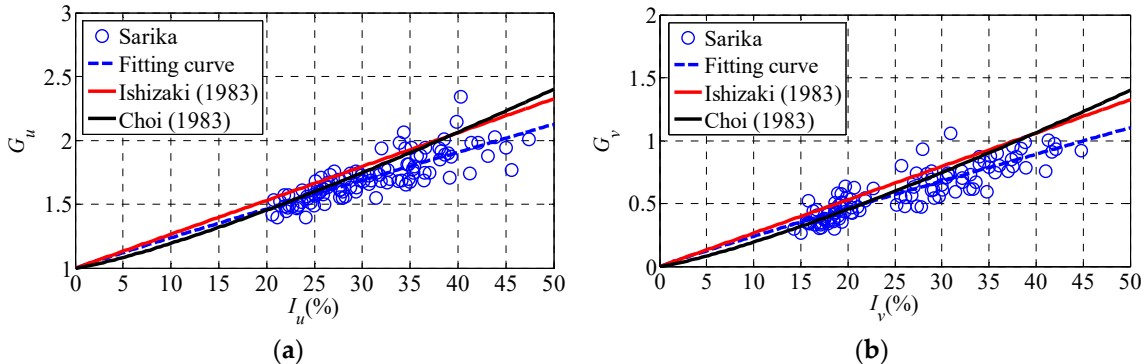

**Figure 10.** The relationship between gust factor and turbulence intensity. (**a**) Longitudinal gust factors; (**b**) Lateral gust factors.

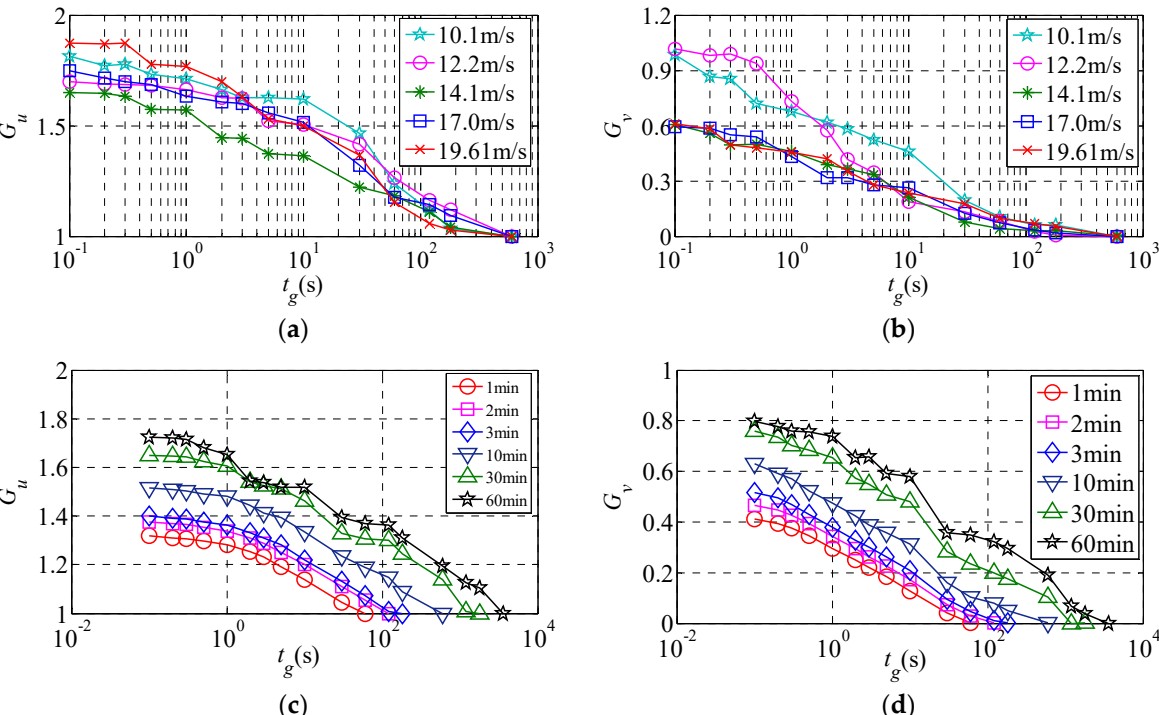

**Figure 11.** The variation of gust factors obtained at different sample duration and gust duration. (**a**) $G_u$ with respect to $t_g$; (**b**) $G_v$ with respect to $t_g$; (**c**) $G_u$ with respect to $T$; (**d**) $G_v$ with respect to $T$.

### 3.5. Power Spectrum of Wind Speeds

The power spectrum of wind speeds can be expressed by dimensionless power spectral density function $fS_u(f)/\sigma_u^2$, and $S_u(f)$ is the power spectrum of wind speeds. Many scholars have conducted significant studies on the power spectrum of wind speeds, including Von Kaimal spectrum, Davenport spectrum, Simiu spectrum and Harris spectrum. Tamura et al. [15] and Xu and Zhan [5] proposed that the power spectrum of wind speed generally followed a Von Karman-type spectrum during typhoons. Therefore, Von Karman-type spectrum were selected to compare with the measured power spectra of wind speeds. The longitudinal and lateral power spectrum of wind speeds are predicted by Von Karman-type spectrum as follows:

$$\frac{fS_u(f)}{\sigma_u^2} = \frac{4X_u}{(1 + 70.8X_u^2)^2} \quad X_u = \frac{fL_u^x}{U} \tag{13}$$

$$\frac{fS_v(f)}{\sigma_v^2} = \frac{4X_v(1 + 755.2X_v^2)}{(1 + 283.2X_v^2)^{11/6}} \quad X_v = \frac{fL_v^x}{U} \tag{14}$$

where $\sigma_u$ and $\sigma_v$ are the standard deviations of fluctuating wind in the longitudinal and lateral direction, respectively; and $L_u^x$ and $L_v^x$ are the integral scales of fluctuating wind in longitudinal and lateral directions, respectively. The values $X_u$ and $X_v$ are the dimensionless frequency of fluctuating wind in longitudinal and lateral directions, respectively. In this study, power spectrum of wind speeds of sample duration at 10-min mean wind speed of 9.09 m/s, 15.14 m/s, and 20.02 m/s are analyzed. Figure 12a–d shows that the Von Karman-type spectrum was consistent with the predicted value of the measured fluctuating wind power spectrum in longitudinal and lateral directions. The Von Karman-type spectrum has a slightly higher prediction of the longitudinal power spectrum of wind speeds when $X_u$ in the range of $10^{-2}$–$10^{-1}$. Moreover, the measured longitudinal power spectrum of wind speeds decays faster than the lateral power spectrum of wind speeds, and the Von Karman-type spectrum has a slightly higher prediction of the longitudinal power spectrum of wind speeds when $X_u$ is in the range of $10^{-1}$~$10^1$.

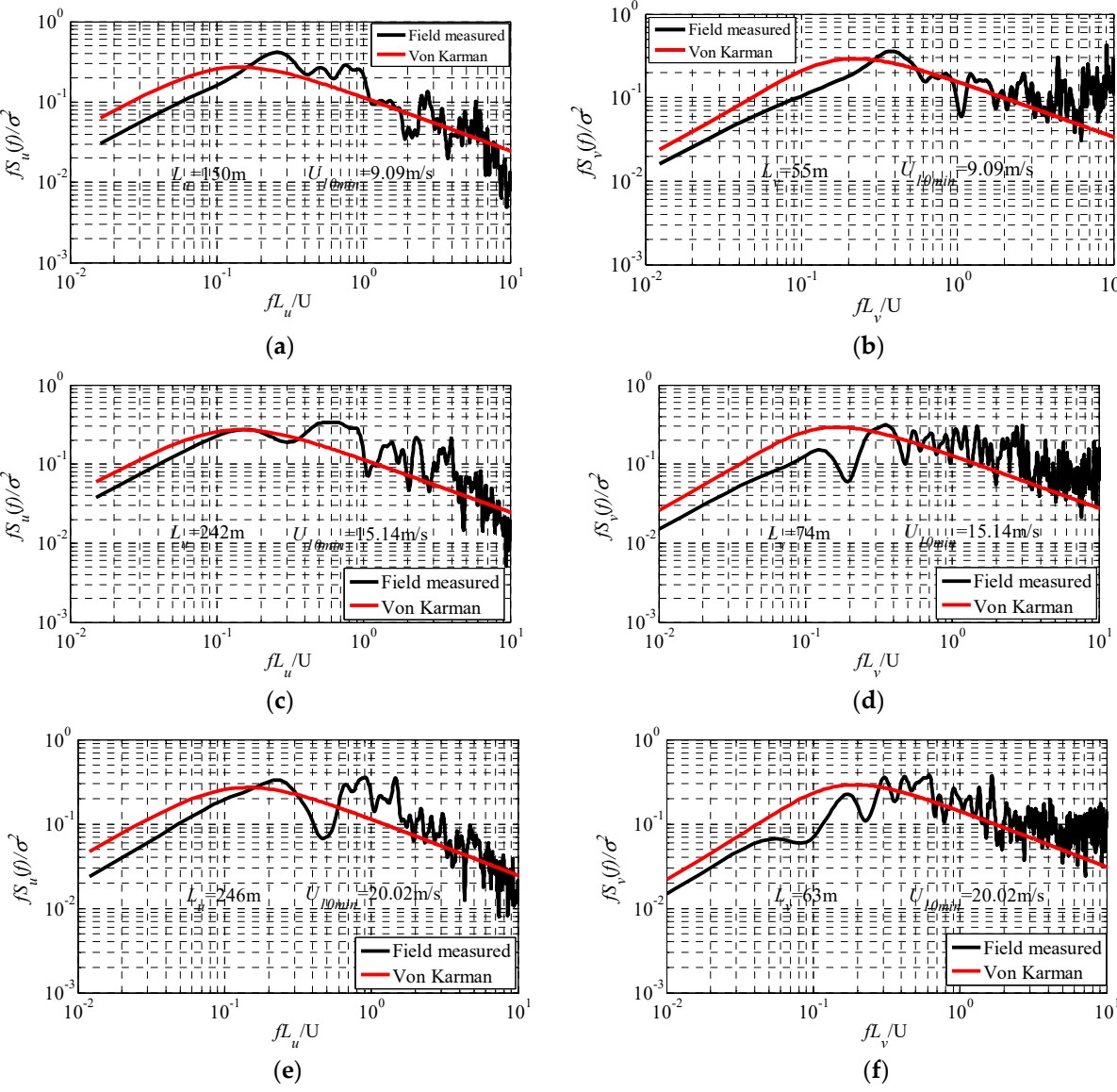

**Figure 12.** Power spectrum of wind speed during Typhoon Sarika. (**a**) $U_{10min} = 9.09$ m/s; $L_u = 150$ m; (**b**) $U_{10min} = 9.09$ m/s; $L_v = 55$ m; (**c**) $U_{10min} = 15.14$ m/s; $L_u = 242$ m; (**d**) $U_{10min} = 15.14$ m/s; $L_v = 74$ m; (**e**) $U_{10min} = 20.02$ m/s; $L_u = 246$ m; (**f**) $U_{10min} = 20.02$ m/s; $L_v = 63$ m.

### 3.6. Time-Frequency Analysis of Non-Stationary Samples of Fluctuating Wind Speed

Based on the Fourier transform, the Wavelet transform adopts translatable time and scalable scale, and establishes a perfect time-frequency domain analysis system. In time-frequency domain analysis, the traditional Fourier transform is only physically meaningful for the strict stationary process, and the spectrum obtained by the Fourier transform is only related to the frequency while neglecting time domain characteristics. Besides, the improved short-time Fourier transform only exhibits the limited ability in the time-frequency domain analysis for its limitation in unchangeable scale, thus it has not been widely used. However, the Wavelet transform shows excellent performances in changeable scale and harmonious frequency and time domain. It can obtain both high time resolution in high frequency region and high frequency resolution in low frequency region; thus, the wavelet transform has favorable application in the analysis of non-stationary signals. Because the scale function of the wavelet transform is inversely proportional to the frequency, its wavelet transform coefficients also reflect the distribution of signal energy in both time and frequency domain to a certain extent.

Five 1-min subsamples (No. 1, No. 2, No. 3, No. 4 and No. 5) in the non-stationary samples are selected for wavelet transform. Figure 13 presents the three-dimensional images of five 1-min subsamples in time-frequency domain and the clear comparison of the energy distribution of fluctuating wind speed in time-domain coordinates, which shows that the amplitude of wavelet coefficients fluctuates greatly in time-domain coordinates, indicating that the larger the wind speed, the larger the amplitude of wavelet coefficients in time-domain. Figure 13 also illustrates that the energy of the wind speed sample gradually decreases in the frequency domain coordinates. After converting the frequency coordinates into dimensionless frequency coordinates according to Equations (13) and (14), the corresponding dimensionless frequency at the peak wavelet coefficients of the non-stationary pulsating wind concentrates basically between 0.1 and 0.2.

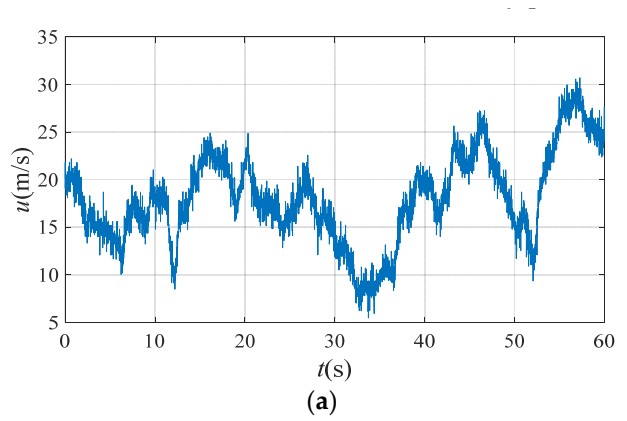

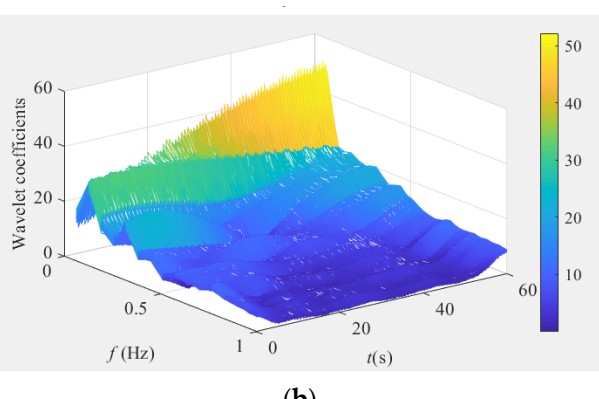

(a)

(b)

**Figure 13.** *Cont.*

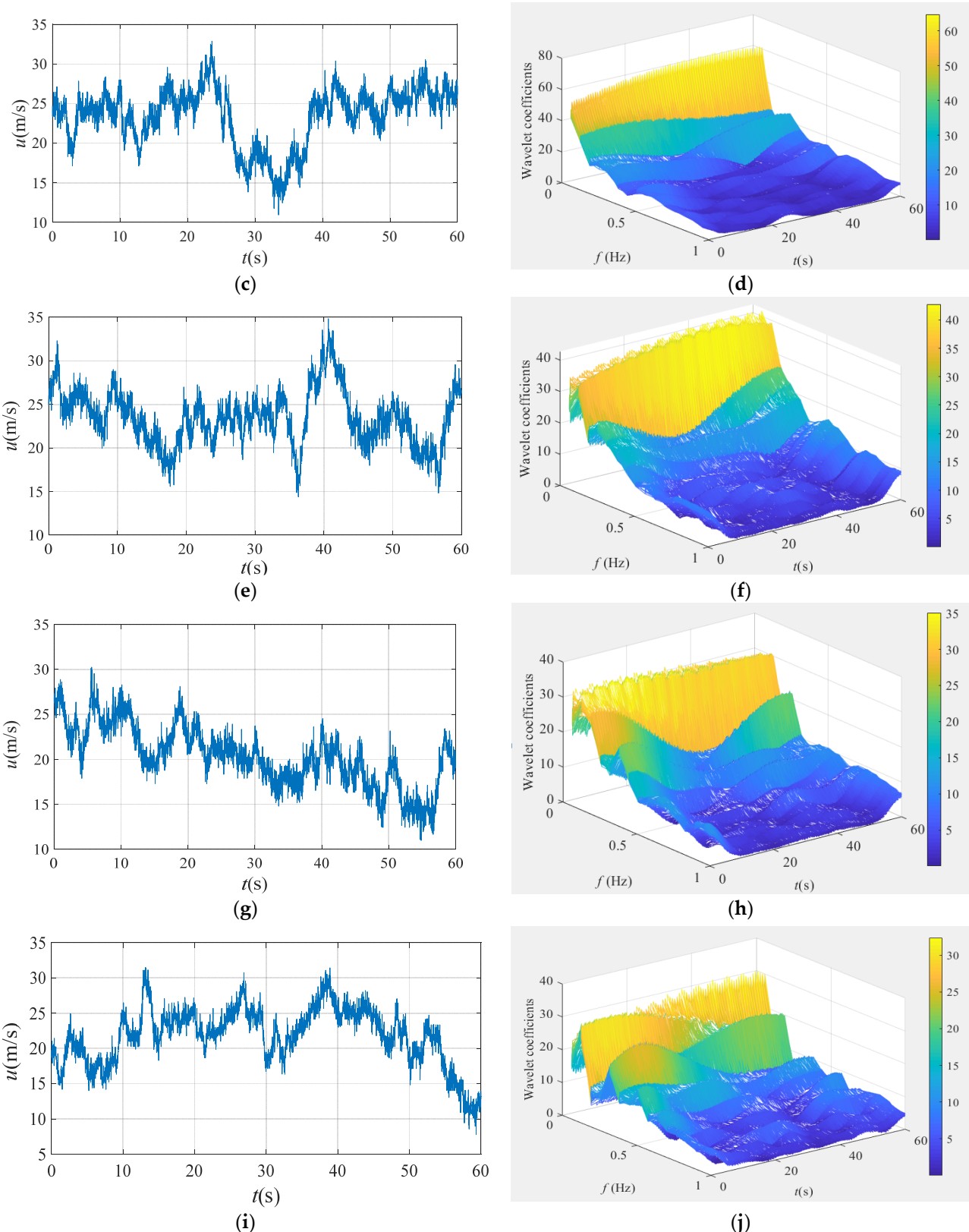

**Figure 13.** Wavelet time-frequency diagram of five 1-min subsamples. (**a**) Wind speed sample of No. 1; (**b**) Wavelet time-frequency diagram of No. 1; (**c**) Wind speed sample of No. 2; (**d**) Wavelet time-frequency diagram of No. 2; (**e**) Wind speed sample of No. 3; (**f**) Wavelet time-frequency diagram of No. 3; (**g**) Wind speed sample of No. 4; (**h**) Wavelet time-frequency diagram of No. 4; (**i**) Wind speed sample of No. 5; (**j**) Wavelet time-frequency diagram of No. 5.

### 3.7. The Measured Probability Density Characteristics of Typhoon Sarika

The distribution of fluctuating wind is often assumed to follow the Gauss distribution in the current methods to predict the dynamic response of structures under the influences of seasonal trade winds:

$$P(u) = \frac{1}{\sqrt{2\pi}\sigma_u} e^{-\frac{u^2}{2\sigma_u^2}} \tag{15}$$

To study whether the probability density function of a typhoon follows the Gaussian distribution, the measured wind field data is analyzed systematically, and the resulting function is compared with the Gaussian distribution. The skewness coefficient (*S*) and kurtosis coefficient (*K*) are vital parameters to judge whether the measured time-history data accord with the Gaussian distribution. Provided that *S* = 0 and *K* = 3 accord with the Gaussian distribution. The formulas for the calculating skewness coefficient and kurtosis coefficient are as follows:

$$S = \frac{1}{n}\sum_{j=1}^{n}\left[\frac{C_{ai}(t_j) - C_{ai,mean}}{C_{ai,std}}\right]^3 \tag{16}$$

$$K = \frac{1}{n}\sum_{j=1}^{n}\left[\frac{C_{ai}(t_j) - C_{ai,mean}}{C_{ai,std}}\right]^4 \tag{17}$$

In Equations (16) and (17), $n$ is the sample size, $C_{ai,mean}$ is the sample mean, and $C_{ai,std}$ is the sample standard deviation. This paper discusses whether the probability density of wind speed in the whole process of typhoon landing conforms to Gaussian distribution. Figure 14 shows the probability density of total samples in the entire landing process of Typhoon Sarika. The skewness coefficient is 0.1787 and the kurtosis coefficient is 2.7572 during the landing process of Sarika. Criteria of $|S| < 0.5$ and $2.0 < |K| < 4.0$ are used for judging whether the probability density of the sample follow a Gaussian distribution. Therefore, the wind speed samples follow the Gaussian distribution during the landing process of Typhoon Sarika.

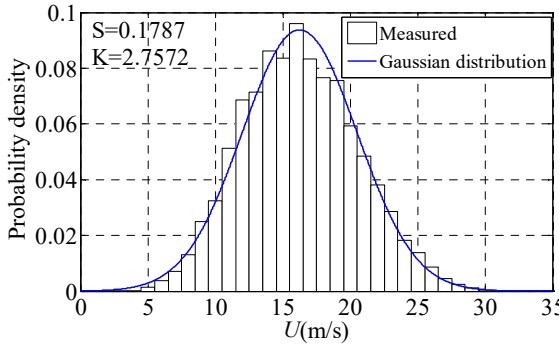

**Figure 14.** Probability density of total wind speed samples in the whole landing process of Typhoon Sarika.

## 4. Conclusions

Wind-field characteristics are monitored synchronously during the passage of Typhoon Sarika in Haikou. A large number of samples covering different wind directions are obtained. The mean wind and fluctuating wind characteristics atop the building under the action of typhoon Sarika are counted. The main conclusions are as follows:

(1) Turbulence intensity decreases with the increasing wind speed and remains slight changes under high wind speed, and the ratios between longitudinal and lateral turbulence intensities are basically within the range of 1.0~2.0. The measured longitudinal and lateral turbulence integral scales increase with the increasing mean wind speed

and S.D of fluctuating wind. The ratio of the turbulence scale between the turbulence components $L_u^x:L_v^x$ increases with the increasing mean wind speed and the increasing rate of longitudinal turbulence integral scales is more significant than that of lateral turbulence integral scales.

(2) The gust factor is related to turbulence intensity, sample duration, and gust duration; the gust factor increases as turbulence intensity increases; and the findings show that the relationship between turbulence intensity and gust factor is approximate to linear. The gust factor increases with the rise of sample duration, while it decreases with the increasing gust averaging time.

(3) The Von Karman-type spectrum has a slightly higher prediction of the longitudinal power spectrum of wind speeds when the dimensionless frequency is in the range of $10^{-2}\sim10^{-1}$. Moreover, the measured longitudinal power spectrum of wind speed decays faster than the lateral power spectrum, and the Von Karman-type spectrum has a slightly higher prediction of the longitudinal power spectrum of wind speeds when the dimensionless frequency $X$ is in the range of $10^{-1}\sim10^1$.

(4) The time-frequency analysis of non-stationary samples of fluctuating wind speed are conducted by wavelet transform, which shows that the amplitude of wavelet coefficients fluctuates greatly in time-domain coordinates, indicating that the higher the wind speed, the larger the amplitude of wavelet coefficients in time-domain.

**Author Contributions:** J.H. contributed to the overall study design, analysis, and writing of the manuscript. Z.L. provided technical support and supervision. Z.Z. provided technical support and modification-polis. All authors have read and agreed to the published version of the manuscript.

**Funding:** This research was financially supported by the Natural Science Foundation of Hunan Province, China (Grant No. 2020JJ5205), Scientific Research Project of Hunan Provincial Department of Education, China (Grant No. 21B0730) and the National Natural Science Foundation of China (Grant No. 91215302).

**Data Availability Statement:** Not applicable.

**Conflicts of Interest:** The authors declare no conflict of interest.

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
