# Peer review of "Full-Scale Measurements of Wind Characteristics on a High-Rise Building during Typhoon Sarika"

_applsci, doi:10.3390/app12010324_

Round 1

Reviewer 1 Report

This paper conducted an in-situ measurement of wind characteristics at the top of a high-rise building during the passage of Typhoon Sarika. The paper is well written and provides new insights into the impact of extreme events on the microclimate condition in urban areas. An updated literature review needs to be provided, particularly in relation to wind gust calculation in extreme events.

Author Response

Comment:This paper conducted an in-situ measurement of wind characteristics at the top of a high-rise building during the passage of Typhoon Sarika. The paper is well written and provides new insights into the impact of extreme events on the microclimate condition in urban areas. An updated literature review needs to be provided, particularly in relation to wind gust calculation in extreme events.

Response:  Thank you for your comments and affirmation. We added an updated literature review in relation to wind gust calculation in extreme events  in the introduction of the revision manuscript.

“Cao et al. (2015) studied the gust factor related to Typhoon Maemi and showed that typhoon gust factors have no significant difference from those associated with non-typhoon winds, and can be described using models developed for standard neutral boundary layer flow conditions. He et al. (2021)[8] investigated the gust factors during Super Typhoon Hato in the South China sea, the results show that the gust factor initially increased with increasing wind speed and plateaued at wind speed between 30 and 50m/s before exhibiting a decreasing trend until 70m/s. Fang et al. (2019) investigated the gust characteristics of near-ground typhoon based on 14 sets of records observed by 4 meteorological stations during 10 tyhpoons, the results show that the off-sea gust factor at 10 m is 8% higher than the over-land value while other heights show little difference. ”

Reviewer 2 Report

In this study, full-scale measurement of wind characteristics (e.g., mean wind speed and turbulence intensity) atop a high-rise building was conducted during the passage of Typhoon Sarika which are critical factors for wind-resistant design of the wind-sensitive infrastructures. However, the field-measured wind characteristics from different observation stations corresponding to various typhoons are quite scattered and hard-to-reach unified conclusions to guide the wind-resistance design of buildings and structures in the cyclone-prone regions. The inconsistent wind characteristics may be attributed to the complexity of turbulence driven mechanisms, e.g., shear, convection, rotation, blocking and sheltering effects at the boundary layer, and also the interactive motions of multi-scale eddies in the flow fields of tropical cyclones.

Considering the fact that only one event was measured (i.e., Typhoon Sarika), it is very hard to draw general conclusions. Therefore, the authors need to ‘standardize’ their wind characteristics to be able to get important conclusions. In addition, the standardized results need to be compared with the corresponding recommendations in standard design codes (e.g. China wind codes or the American Society of Civil Engineers …) to assess their applicability for such conditions. For instance, in the China wind codes, wind characteristics are specified over standard terrain with roughness length of 0.03 m, averaging time of 10 min for mean wind and duration time of 3 s for gusty wind at 10 m height. Accordingly, it is essential to convert the turbulence characteristics obtained from the measurements to the standard terrain and investigate the wind nature in a unified framework. The “standardized” wind characteristics due to their universality could be useful in instructing the structural design in typhoon-prone regions.

In addition, the authors need to explicitly highlight the data quality control and corrections to correct their measurements (not addressed in their paper). This is important, because typhoons are characterized by strong winds accompanied by torrential rain which affect the measurements. Therefore, the data quality procedure is a necessary step before analyzing the wind characteristics.

The authors need to review their paper very carefully to correct several grammatical errors and sentences. For instance, (not an exhaustive list) these are few sentences that need to be corrected:

Line 82: On October 19, Sarika deteriorated 81 rapidly as made its final landfall over

Line 98: It can considere as Category-A in Chinese Code corresponds to category-0 and I in the European Code, and Category-B, Category-C and Category-D correspond to the Category-II, Category-III and Category-IV, respectively.

Line 117: Fig. 3 shows the installation location of Model 05103V anemometer, wind field at a height of 115m is recorded by …

Line 193: Fig. 7 shows the 10-min variations longitudinal and lateral turbulence intensities …

Line 246: Is it Equation 8 or 2?

Line 277: The gust factor … is defines as the

Line 397: The fluctuating wind is often assumed as the Gauss distribution in

Author Response

Comment:In this study, full-scale measurement of wind characteristics (e.g., mean wind speed and turbulence intensity) atop a high-rise building was conducted during the passage of Typhoon Sarika which are critical factors for wind-resistant design of the wind-sensitive infrastructures. However, the field-measured wind characteristics from different observation stations corresponding to various typhoons are quite scattered and hard-to-reach unified conclusions to guide the wind-resistance design of buildings and structures in the cyclone-prone regions. The inconsistent wind characteristics may be attributed to the complexity of turbulence driven mechanisms, e.g., shear, convection, rotation, blocking and sheltering effects at the boundary layer, and also the interactive motions of multi-scale eddies in the flow fields of tropical cyclones.

Response: Thank you for your comments.  I agree with you that the field-measured wind characteristics from different observation stations corresponding to various typhoons are quite scattered. Due to the uncertainty of typhoon landing path and the limitation of test conditions, we can only test the typhoon wind field over high-rise buildings at fixed locations. The wind characteristics will be different if the measuring points are located in different wind circles of the typhoon. In the follow-up study, the monitoring and comparison of the measured data of wind fields at different measuring points will be strengthened.

Comment:Considering the fact that only one event was measured (i.e., Typhoon Sarika), it is very hard to draw general conclusions. Therefore, the authors need to ‘standardize’ their wind characteristics to be able to get important conclusions. In addition, the standardized results need to be compared with the corresponding recommendations in standard design codes (e.g. China wind codes or the American Society of Civil Engineers …) to assess their applicability for such conditions. For instance, in the China wind codes, wind characteristics are specified over standard terrain with roughness length of 0.03 m, averaging time of 10 min for mean wind and duration time of 3 s for gusty wind at 10 m height. Accordingly, it is essential to convert the turbulence characteristics obtained from the measurements to the standard terrain and investigate the wind nature in a unified framework. The “standardized” wind characteristics due to their universality could be useful in instructing the structural design in typhoon-prone regions.

Response: Thank you for your comments. Due to the obvious difference between the characteristics of urban high-altitude wind field under the action of typhoon and normal wind and monsoon, especially the interference of urban high-altitude buildings will lead to a large difference between the typhoon wind profile and several types of wind profiles specified in the code, and the urban geomorphic types are complex. The geomorphic categories in different incoming directions of measured places are inconsistent, and the instantaneous wind speed and wind direction of typhoon are fluctuating, If the measured high-altitude wind field is converted to the wind field at 10m height, there will be obvious deviation, so it is often difficult to standardize the measured wind characteristics of typhoon according to the wind tunnel test conditions. In the manuscript, the measured height turbulence intensity is calculated and discussed according to the standard formula, Eq. (7) can be used to calculate the turbulence intensity at a height of 115m with respect to four categories of terrain roughness are 8.95%, 9.71%, 13.4% and 18.8%, respectively. The measured turbulence intensity with a 10-min interval is 15%-40% during Typhoon Sarika, which indicates that the measured wind field characteristic is greatly subject to the interference of the ambient condition, thus the category of terrain roughness for most of the inflow wind is D. The measured turbulence intensity is 10% along the critical wind directions of 160° is close to category A of terrain roughness.

Comment:In addition, the authors need to explicitly highlight the data quality control and corrections to correct their measurements (not addressed in their paper). This is important, because typhoons are characterized by strong winds accompanied by torrential rain which affect the measurements. Therefore, the data quality procedure is a necessary step before analyzing the wind characteristics.

Response: Thank you for your comments. We added the data quality procedure in the revision manuscript.

“Firstly, in order to prevent high-frequency signal interference in the process of typhoon wind field data acquisition, it is necessary to carry out band-pass filtering on the measured wind field data before analyzing the data. Because the frequency domain distribution of wind power spectrum is concentrated in the range of 0~12Hz, the band-pass filtering range used in this paper is 0~12Hz. Secondly, the measured process is often accompanied by power frequency, frequency doubling and irregular random interference noise signals, resulting in obvious burr points in the measured data. A large peak data occasionally appears in the time history data. If it is not removed in time, it will have a great impact on the mean value and mean square deviation of the data. Therefore, after band-pass filtering the measured wind field data, the data with burr points are interpolated and replaced by the five point cubic method to eliminate the error caused by burr points. ”

 The authors need to review their paper very carefully to correct several grammatical errors and sentences. For instance, (not an exhaustive list) these are few sentences that need to be corrected:

Comment:Line 82: On October 19, Sarika deteriorated rapidly as made its final landfall over

Response: Thank you for your comments.  We have corrected the grammar of the sentence in the revision manuscript.

“The typhoon Sarika, which decays rapidly on  October 19 at 17:00, make its final landfall over the coastline and border of Vietnam and China”.

Comment:Line 98: It can considere as Category-A in Chinese Code corresponds to category-0 and I in the European Code, and Category-B, Category-C and Category-D correspond to the Category-II, Category-III and Category-IV, respectively.

Response: Thank you for your comments.  We have corrected the word of this sentence in the revision manuscript.

“It can consider as Category-A in Chinese Code corresponds to category-0 and I in the European Code, and Category-B, Category-C and Category-D correspond to the Category-II, Category-III and Category-IV, respectively. ”

 Comment:Line 117: Fig. 3 shows the installation location of Model 05103V anemometer, wind field at a height of 115m is recorded by …

Response: Thank you for your comments.We have corrected the sentence in the revision manuscript.

“Fig. 3 shows the installation location of Model 05103V anemometer at a height of 115m, the instantaneous wind speed is recorded by the mechanical anemometer with the maximum measurement capacity of 100.00 m/s. ”

 Comment:Line 193: Fig. 7 shows the 10-min variations longitudinal and lateral turbulence intensities …

Response: Thank you for your comments.We have corrected the sentence in the revision manuscript.

“ Fig. 7 shows the relationship of the 10-min variations longitudinal and lateral turbulence intensities (and ) with respect to mean wind speeds and directions (). ”

 Comment:Line 246: Is it Equation 8 or 2?

Response: Thank you for your comments.We have corrected the sentence in the revision manuscript.

“Eq. (8) shows that the turbulence integral scale is closely related to mean wind speed and the standard deviation of S.D of fluctuating wind.”

 Line 277: The gust factor … is defines as the

Response: Thank you for your comments.We have corrected the sentence in the revision manuscript.

“The gust factor  is defined as the ratio of the gust speed  in gust duration  to the 10-min mean wind speed , which can be written as: ”.

 Line 397: The fluctuating wind is often assumed as the Gauss distribution in

Response: Thank you for your comments.We have corrected the sentence in the revision manuscript.

“The distribution of fluctuating wind is often assumed to follow the Gauss distribution in the current methods to predict the dynamic response of structures under the influences of seasonal trade winds: ”

Reviewer 3 Report

The paper topic is very interesting and promising. I shall thank the author's for their research before anything.

However,I need to introduce some suggestions for them to increase the quality and presentation of the paper.

1- introduction section is very long please summerize and highlight the main topic of the research on the paper.

2- table 1 isn't appropriate in is current forum in the paper.

3-figure 1 is very wide in location description pease focus more or add other shape with more focus.

4- colors of figure 3 very dark change it to more brighter.

5- add more new citations in the manuscript topic.

6- please summerize the conclusion ad don't repeat grades from results as it is.

Author Response

The paper topic is very interesting and promising. I shall thank the author's for their research before anything.

 However,I need to introduce some suggestions for them to increase the quality and presentation of the paper.

Comment:1- introduction section is very long please summerize and highlight the main topic of the research on the paper.

Response: Thank you for your comments. In the introduction section, we added the following contents to summarize and highlight the main topic of the research.

“Based on the field measured data of the wind field on the roof under the influence of five typhoons, the characteristics of average wind and fluctuating wind during typhoon landing are analyzed, mainly including turbulence intensity, turbulence integral scale, gust factor, probability density function and fluctuating wind speed spectrum, as well as the internal relationship between these parameters and mean wind speed and wind direction, This paper reveals the relationship between gust factor and basic time interval and gust duration, puts forward the empirical formula for predicting the relationship between gust factor and turbulence intensity, and discusses the characteristics of fluctuating wind speed spectrum at different stages of typhoon landing. The measured building is elected to conduct wind-resistant field measurement and analyze the impact of different inflow wind field types on the wind-induced response of high-rise buildings due to the diverse geomorphic characteristics of the building in different directions.”

 Comment:2- table 1 isn't appropriate in is current forum in the paper.

Response: Thank you for your comments. We deleted Table 1 in the corresponding section.

Comment:3-figure 1 is very wide in location description pease focus more or add other shape with more focus.

Response: Thank you for your comments. We have replaced Fig.1 in the revision manuscript.

Comment:4-colors of figure 3 very dark change it to more brighter.

Response: Thank you for your comments.  We corrected the brightness of Fig. 3.

Comment:5add more new citations in the manuscript topic.

Response: Thank you for your comments. We added the following two latest literatures.

“Cao et al. (2015) studied the gust factor related to Typhoon Maemi and showed that typhoon gust factors have no significant difference from those associated with non-typhoon winds, and can be described using models developed for standard neutral boundary layer flow conditions. He et al. (2021)[8] investigated the gust factors during Super Typhoon Hato in the South China sea, the results show that the gust factor initially increased with increasing wind speed and plateaued at wind speed between 30 and 50m/s before exhibiting a decreasing trend until 70m/s. Fang et al. (2019) investigated the gust characteristics of near-ground typhoon based on 14 sets of records observed by 4 meteorological stations during 10 tyhpoons, the results show that the off-sea gust factor at 10 m is 8% higher than the over-land value while other heights show little difference. ”

Comment:6-please summerize the conclusion ad don't repeat grades from results as it is.

Response: Thank you for your comments. We revised the content of the conclusion to avoid repeated discussion as far as possible, Make the summary prominent and concise.

Round 2

Reviewer 2 Report

Thanks you for your detailed and thoughtful responses.